# Unveiling Delay Effects in Traffic Forecasting: A Perspective from Spatial-Temporal Delay Differential Equations

Submission Id: 2279

## ABSTRACT

Traffic flow forecasting is a fundamental research issue for transportation planning and management, which serves as a canonical and typical example of spatial-temporal predictions. In recent years, Graph Neural Networks (GNNs) and Recurrent Neural Networks (RNNs) have achieved great success in capturing spatial-temporal correlations for traffic flow forecasting. Yet, two non-ignorable issues haven't been well solved: 1) The message passing in GNNs is immediate, while in reality the spatial message interactions among neighboring nodes can be delayed. The change of traffic flow at one node will take several minutes, i.e., *time delay*, to influence its connected neighbors. 2) Traffic conditions undergo continuous changes. The prediction frequency for traffic flow forecasting may vary based on specific scenario requirements. Most existing discretized models require retraining for each prediction horizon, restricting their applicability. To tackle the above issues, we propose a neural **S**patial-**T**emporal **D**elay **D**ifferential **E**quation model, namely STDDE. It includes both delay effects and continuity into a unified delay differential equation framework, which explicitly models the time delay in spatial information propagation. Furthermore, theoretical proofs are provided to show its stability. Then we design a learnable traffic-graph time-delay estimator, which utilizes the continuity of the hidden states to achieve the gradient backward process. Finally, we propose a continuous output module, allowing us to accurately predict traffic flow at various frequencies, which provides more flexibility and adaptability to different scenarios. Extensive experiments show the superiority of the proposed STDDE along with competitive computational efficiency. Moreover, both quantitative and qualitative experiments are conducted to validate the concept of a delay-aware module. Also, the flexibility validation shows the effectiveness of the continuous output module.

## KEYWORDS

deep graph learning, differential equation, traffic network, traffic flow prediction, continuous systems

## 1 INTRODUCTION

Traffic forecasting is a fundamental research problem of Intelligent Transportation Systems (ITS) [29, 30, 41], which affects a variety of smart city applications [8, 35], such as trip planning [1, 8, 11]

*Conference'17, July 2017, Washington, DC, USA*
© 2023 Association for Computing Machinery.
ACM ISBN 978-x-xxxx-xxxx-x/YY/MM...$15.00
https://doi.org/10.1145/nnnnnnn.nnnnnnn

accident prediction [17, 19], and urban management [5, 42]. Traffic flow forecasting aims to predict the future traffic flow based on historical data and underlying traffic networks.

Traffic flow forecasting is a challenging task due to the inherent spatial-temporal dependencies. Benefiting from the flourishing of deep learning, a large number of deep models have been proposed for traffic forecasting. In the temporal dimension, RNN-based models and their variants [33] occupy the mainstream status, and temporal convolution networks [27] have also attracted much attention due to their superior computation efficiency. In the spatial dimension, considering that most traffic networks are non-Euclidean other than grid-partitioned, GNN-based methods [25, 41] beat CNN-based ones [43] and become predominant owing to their strong ability to deal with graph-structured data. Extensive works combine the spatial module and the temporal module to achieve significant improvements, among which STGCN [41] and DCRNN [25] and DSTAGN [22] are the representative.

Nevertheless, previous works prove the following shortcomings, **(1) The delays in the graph signal propagation process are overlooked.** When an incident occurs at specific nodes, the influence will take several minutes (i.e. *time delay*) to propagate to their neighboring nodes. However, the delay effect is largely neglected in existing spatial-temporal traffic forecasting issues. Time series models [4, 13], such as RNN and GRU, are capable of modeling scenarios in which the delay remains consistent across all nodes and timestamps. In contrast, the practice is quite the opposite, **time-delays vary significantly at different nodes and timestamps**, as illustrated in Fig. 1(c). It shows the time-delay distribution among neighbors in the PEMS03 dataset. Therefore, a separate module is required to characterize and model these variations. As shown in Fig. 1(a) and 1(b), general GNNs propagate the suddenly changed message indistinguishably based on the adjacency relation, leading to a sub-optimal prediction ahead of the ground truth. Thus it is urgent to involve delay effects in spatial-temporal traffic forecasting.

**(2) The inherent continuity in traffic system is not well-explored.** Existing methods mainly utilize RNNs [33] or TCNs [27], which accept discrete observations as input, to capture the temporal dependencies. These methods are limited in terms of flexibility and applicability. Specifically, for the same traffic system, the required prediction horizon and resolution may vary across different applied scenarios, and the model needs to be retrained for each specific demand. Also, traffic data is notable for its inherent sparsity [26, 44]. This sparsity arises due to the limited availability of traffic sensors, particularly in extensive road networks. For example, the sampling precision of sensors deployed within the traffic network may be at a 10-min interval. However, during the prediction phase, we aspire to attain finer-grained forecasting precision to enable rapid responses to events, such as travel time planning [1] or traffic emergency management [44].

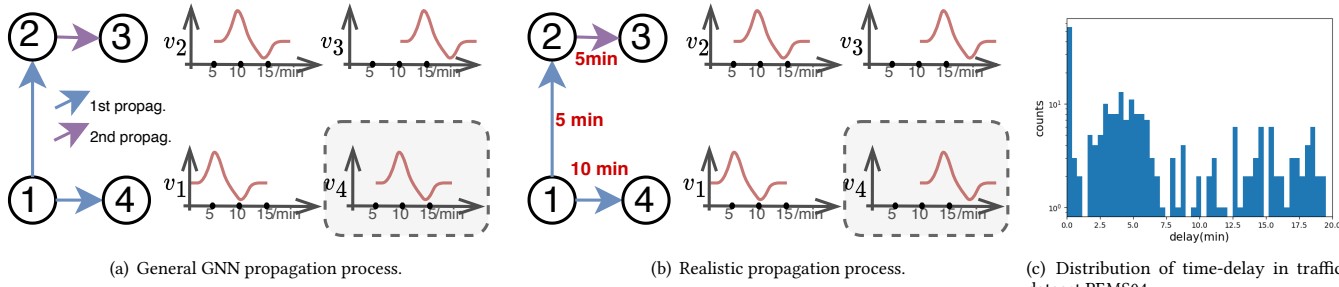

(a) General GNN propagation process.

(b) Realistic propagation process.

(c) Distribution of time-delay in traffic dataset PEMS04.

Figure 1: (a) and (b) show the comparison of spatial-temporal signal propagation between general GNNs and realistic conditions. Node 2 and node 4 receive the same update information simultaneously in graph propagation, while they do not in the realistic scene. Fig. (c) shows the distribution of delay values in the real-world traffic network, which are computed based on the max-cross-correlation method[2].

To tackle above mentioned issues, we propose a neural Spatial-Temporal Delay Differential Equation model (STDDE). In contrast to existing methods, STDDE presents an innovative paradigm for spatial-temporal traffic analysis by addressing the aforementioned two challenges. STDDE explicitly captures and leverages delayed spatial interactions among neighboring nodes. Furthermore, it models spatial-temporal evolution signals from a continuous perspective, departing from traditional recurrent approaches. Specifically, for each node, we first estimate the delay values among its neighbors to build a delayed connected graph. Then we combine the specific historical hidden states of its own and its neighbors to effectively integrate spatial and temporal information by using a Delay-aware Differential Equation (DDE). Then we theoretically prove the proposed delay differential equation is asymptotically stable. We conduct experiments on six popular used real-world traffic datasets. The results demonstrate that our model outperforms state-of-the-art models while maintaining competitive computational efficiency. Quantitative and qualitative experiments are conducted to validate the effectiveness of the delay-aware module. Additionally, the flexibility validation confirms the effectiveness of the continuous output module.

The main contributions of this work are summarized as follows:

- We propose a **S**patial-**T**emporal **D**elay **D**ifferential **E**quation model, namely STDDE, which includes both delay effects and continuity into a unified delay differential equation framework, which explicitly models the time delay in spatial information propagation.
- We design a learnable traffic-graph time-delay estimator, which utilizes the continuity of the hidden states to achieve the gradient backward process.
- We propose a continuous output module, allowing us to accurately predict traffic flow at various frequencies, which provides more flexibility and adaptability to different scenarios.
- We conduct experiments on six popular used datasets, in which results show that our model outperforms the SOTAs

and exhibits competitive computational efficiency. Moreover, both quantitative and qualitative experiments are conducted to validate the concept of delay-aware module. Also, the flexibility validation shows the effectiveness of the continuous output module.

## 2 RELATED WORK

### 2.1 Traffic Flow Forecasting

A large body of research has been conducted on traffic flow forecasting in recent years. Traffic flow forecasting can be viewed as a spatial-temporal forecasting task leveraging spatial-temporal data collected by various sensors to predict future traffic conditions. In recent years, deep learning methods have dominated traffic flow forecasting issues, due to their superior ability to model complex spatial-temporal correlations. The models combining graph neural networks (GNN) [21] and recurrent neural networks (RNN) [33] are the representative. Specifically, DCRNN [24] views the traffic flow as a diffusion process on a directed graph and utilizes GRU to capture the temporal features. STGCN [41] utilizes graph convolution and 1D convolution to capture spatial dependencies and temporal correlations respectively. ASTGCN[14] introduces an attention mechanism to capture the dynamics of spatial dependencies and temporal correlations. And AGCRN [3] captures node-specific spatial and temporal correlations automatically without a pre-defined graph. [18] proposes a PDFormer model. While PDFormer mentions the concept of delay, its core mechanism involves utilizing attention to the historical time series, rather than explicitly utilizing delay in the propagation process. It implies that it still cannot capture the intricate delay information among graph vertices.

In general, despite their achieved success, all existing works are limited to the spatial-temporal stacking structure and ignore the delay effect, which deviates from the real situation of traffic.

### 2.2 Neural Differential Equations

The neural ordinary differential equation (NODE) [7] was first proposed as a continuous version of residual neural networks (ResNet) [39]. Due to its apparent suitability for dynamics-governed time-series, NODE is soon utilized in the time series analysis, especially

when the input data is irregularly sampled or partially observed [9, 23, 32]. However, the solution of neural ODE is totally determined by the initial condition, which means later arriving data would not exert influence on the equation, this is also why neural ODE is generally applied cooperatively with RNN modules to deal with incoming data. Neural control differential equation (NCDE) [20] solved this problem by constructing a continuous path from discrete input data, and adjusting the evolution trajectory according to the continuous control signal. Another parallel and relevant work is the neural delay differential equation (NDDE) [45]. As emphasized in [10], the flow of NODE cannot represent the systems with the effect of time delay. The emergence of NDDE fills the blank.

Few pioneering works have been conducted in traffic forecasting with a neural differential equation framework. STGODE [12] first utilizes the neural ODE to transform graph convolution into a continuous version to acquire a larger spatial-temporal receptive field. STG-NCDE [8] adopts the neural CDE to deal with irregular-sampled time series. Despite the success have achieved, none of these works take the delay effect into consideration. In this paper, we first extend the NDDE to multi-variable conditions for spatial-temporal modeling and cooperate with NCDE to construct continuous traffic signal evolution.

## 3 PRELIMINARY

### 3.1 Problem Definition

In this paper, we focus on the long-term traffic flow forecasting problem. The traffic network is represented as a graph $\mathcal{G} = (V, E, A)$, where $V$ is the set of $N$ traffic nodes, $E$ is the set of edges, and $A \in \mathbb{R}^{N \times N}$ is an adjacency matrix representing the connectivity of $N$ nodes. The traffic flow is represented as a flow matrix $X \in \mathbb{R}^{T \times N}$, and $X_{tn}$ denotes the traffic flow of node $n$ at time $t$. The goal of traffic flow forecasting is to learn a mapping function $f$ to predict the future $T'$ steps traffic flow given the historical $T$ steps information, which can be formulated as follows,

$$\left[ X_{t-T+1,:}, X_{t-T+2,:}, \cdots, X_{t,:}; \mathcal{G} \right] \xrightarrow[train]{f} \left[ X_{t+1,:}, X_{t+2,:}, \cdots, X_{t+T',:} \right]. \tag{1}$$

Moreover, the model trained at some fixed grain may need to generate a differently-grained prediction to satisfy the complicated real-world needs, which is formulated as follows,

$$\left[ X_{t-T+1,:}, X_{t-T+2,:}, \cdots, X_{t,:}; \mathcal{G} \right] \xrightarrow[infer]{f} \left[ X_{t+dt_1,:}, X_{t+dt_2,:}, \cdots, X_{t+dt_n,:} \right]. \tag{2}$$

where $dt_1, dt_2, \cdots, dt_n$ are **arbitrary** positive numbers.

### 3.2 Neural Differential Equations

*3.2.1 Neural ODE (NODE).* The residual connection structure can be viewed as a discrete manner of Neural Ordinary Differential Equation (NODE). The update of representation $h$ is a special case of the following equation,

$$h_{t+\Delta t} = h_t + \Delta t \cdot f(h_t, \theta_t), \tag{3}$$

with $\Delta t = 1$. Through letting $\Delta t \to 0$ and unifying $\theta_t$ into $\theta$ for parameter efficiency, we get the continuous version,

$$h(T) = h(0) + \int_0^T f\left(h(t), t, \theta\right) \mathrm{d}t, \tag{4}$$

where $h(0)$ is acquired from an input transformation. As the derivative in the ODE is parameterized with a neural network, the above version is named Neural ODE. To achieve memory efficiency, the adjoint sensitivity method is adopted in the backward process [7], which computes the gradients through another ODE rather than step-by-step backpropagation.

*3.2.2 Neural NCDE (NCDE).* To establish connections with input data, NCDE is proposed and formulated as follows,

$$h(T) = h(0) + \int_0^T f(h(t), t, \theta) \mathrm{d}X_t, \tag{5}$$

where the integral is a Riemann–Stieltjes integral [28], and $X_t$ can be viewed as a signal controller in driving the equation evolution process. As Fig. 2 shows, neural CDE eliminates the discontinuity at the data arriving point and renders the whole process continuous in the hidden manifold space.

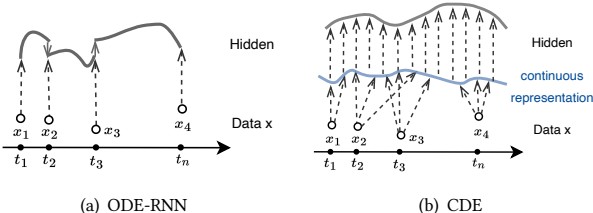

(a) ODE-RNN        (b) CDE

**Figure 2: The workflow comparison of original discrete time-series processing and CDE processing scheme.**

## 4 MODEL: STDDE

Fig. 3 shows the overall framework of our proposed STDDE. It consists of two components. The first component includes both delay effects and continuity into a unified delay differential equation framework, which explicitly models the time delay in spatial information propagation. As the Fig. 3 shows, the hidden state of one node, and the flows evolve in a delay-effect manner, i.e. the hidden state of $v_i$ at $t$ is influenced by the state of $v_j$ at $t - \tau_{ij}$, where $\tau_{ij}$ is the delay from $v_j$ to $v_i$. The second component is the continuous output module, allowing us to accurately predict traffic flow at various frequencies. More details will be demonstrated in the following sections.

### 4.1 Spatial-temporal Delay-aware Neural Differential Equations

*4.1.1 Neural DDE (NDDE).* We first introduce the framework of delay-aware neural differential equations. NDDE introduces the delay effect to improve the precision of signal modeling, in which

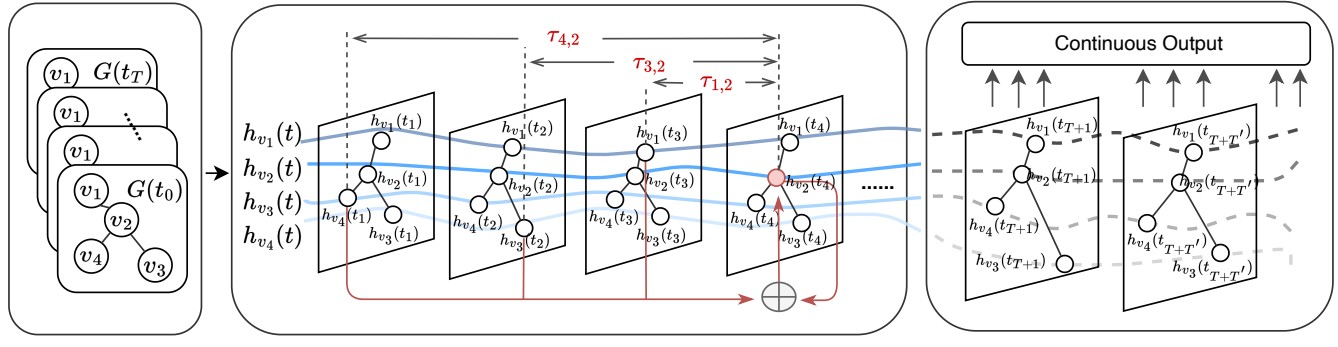

Figure 3: Overview of proposed STGDDE. It consists of two components. The first component includes both delay effects and continuity into a unified delay differential equation framework, which explicitly models the time delay in spatial information propagation. The second component is the continuous output module, allowing us to accurately predict traffic flow at various frequencies.

the evolution process is related to its history,

$$h(t) = \begin{cases} \phi(t), & t \leq 0, \\ h(0) + \int_0^T f(h(t), h(t-\tau), t, \theta)\mathrm{d}t, & t > 0. \end{cases} \quad (6)$$

where $\tau$ is the delay value, and $\phi(t)$ is the history function. The introduction of the delay $\tau$ extends the representation ability of neural ODE and enables modeling a more complex evolution process.

In this paper, we take the GRU [13] as an example to elaborate on the specific derivation of STDDE. Specifically, let $h_t, z_t, g_t$ denote the hidden state, update gate, and update vector respectively, the GRU is defined as follows,

$$z_t = \sigma(W_z h_{t-1} + U_z g_t + b_z), \quad (7)$$
$$h_t = z_t \odot h_{t-1} + (1 - z_t) \odot g_t,$$

where $\sigma$ is the sigmoid activation function, $W_z, U_z$ and $b_z$ are parameters, and $\odot$ denotes element-wise production. By subtracting $h_{t-1}$ from this update equation, we have

$$\Delta h = h_t - h_{t-1} = (1 - z_t) \odot (g_t - h_{t-1}). \quad (8)$$

This naturally leads to the following ODE,

$$\frac{\mathrm{d}h(t)}{\mathrm{d}t} = (1 - z(t)) \odot (g(t) - h(t)). \quad (9)$$

Different from ODE, DDE requires a continuous history function rather than a single point, to serve as the initial stat. A common practice is to set the history function as a time-constant one and approximate it with a multi-layer perception according to the input data,

$$\phi(t) = \text{constant} = \text{MLP}(x), \quad t \leq 0 \quad (10)$$

where $x$ is the input data.

*4.1.2 Incorporating Spatio-temporal Delayed Correlations into NDDE.*
We extend differential equations to the spatial-temporal modeling of the traffic domain. To incorporate spatial-temporal correlations,

we utilize graph neural networks to extract spatial features and view them as update vectors, that is,

$$g_i(t) = c \sum_{j \in \mathcal{N}(i)} \alpha_{ij} f(h_j(t - \tau_{ij})), \quad (11)$$

where $g_i(t)$ is the update vector of node $i$ at time $t$, $\alpha_{ij}$ and $\tau_{ij}$ are the edge weight and delay value between node $v_i$ and $v_j$ respectively, $\mathcal{N}(i)$ denotes the neighbors of node $v_i$, $f$ is a linear transformation, and $c$ is a constant to control the ratio of spatial information. In this formulation, we extend DDE to accommodate multi-variable conditions, to choose the specific history hidden states for node update. Specifically, to update the representation of node $v_i$ at time $t$, we incorporate information from its neighbor $v_j$ at time $t - \tau_{ij}$, taking into account the delayed information propagation with a time delay of $\tau_{ij}$. The graph convolution operation is implemented with DGL package [38] in this work, whose complexity is proportional to the number of edges.

## 4.2 Traffic-Graph Time-Delay Estimator

As shown in Fig. (1), there exist propagation delays in real-world traffic conditions, which are in contrast to the general GNN propagation process. For instance, when a traffic incident transpires in a particular area, it may necessitate several minutes to impact traffic conditions in adjacent regions. To gain more accurate depictions of the time delay, we design two delay estimators to capture the propagation delay between connected nodes.

The direct approach estimates delays among neighboring nodes by maximizing the cross-correlation (MCC) [2] as a pre-processing step. Specifically, given two time series, $x_i$ and $x_j$, where we assume that $x_j$ is influenced by $x_i$, we initially smooth them through interpolation, resulting in $\tilde{x}_i$ and $\tilde{x}_j$. Subsequently, we determine the delay, denoted as $\tau_{ij}$, by identifying the peak of their cross-correlation after shifting, as expressed by the below equation,

$$\tau_{ij} = \arg\max_k \text{corr}(\tilde{x}_i^{\rightarrow k}, \tilde{x}_j), \quad (12)$$

where $\tilde{x}_i^{\to k}$ denotes performing a $k$-step shift to $\tilde{x}_i$, and *corr* is the Pearson correlation function in this paper. We estimate all the delay values in advance through pre-processing based on historical data.

The second approach involves modeling time delay as a learnable pattern. The time delay implicitly reflects external factors associated with the traffic network, including road length, road capacity, and more. Furthermore, the delay itself exhibits inherent variability, for example, longer delays often occur during morning and evening rush hours. In this approach, we assign two learnable delay parameters to each edge: one for peak hours and another for non-peak hours.

Please note that the delay value $\tau$ serves as an indicator to select a historical state in the equation (11). Generally, $\tau$ is considered non-learnable in this context because the model cannot compute the gradient of $\tau$, which is theoretically $\frac{dh(t-\tau)}{d\tau}$. However, thanks to the continuous modeling approach, we can indeed obtain this gradient. As demonstrated in equation (9), the derivative of $h$ with respect to $t$ is well-defined. Consequently, we can incorporate the gradient of $\tau$ in the neural network by explicitly defining the backward computation of $h$ with respect to $\tau$.

## 4.3 State Evolution Controller

One key challenge with DDE is that, once the network parameters are fixed, the dynamic evolution becomes entirely self-contained and does not integrate incoming inputs, leading to the loss of valuable information. To address this issue, we introduce a control signal inspired by Neural CDE, offering a solution to this problem.

Following Neural CDE, we generate a continuous representation from the raw inputs through the natural cubic spline method, which ensures a minimum of two continuous differentiable properties,

$$X(t) = \Phi\left(\{x^0, t_0\}, \{x^1, t_1\}, \cdots, \{x^n, t_n\}\right), \qquad (13)$$

where $X(t)$ is a continuous representation, $\Phi$ denotes the natural cubic spline function, and $x^i$ denotes the input at time $t_i$. Thus we have

$$\frac{dh_i(t)}{dt} = (1 - z_i(t)) \odot (g_i(t) - h_i(t))\tilde{f}\left(\frac{dX_i(t)}{dt}\right), \qquad (14)$$

where $\tilde{f}$ is a transformation function to match dimensions, and $X_i$ is the continuous representation of node $v_i$. The derivative of $X$, denoted as $\frac{dX_i(t)}{dt}$, signifies the trend or fluctuation in traffic flow, constantly influencing the direction of dynamic evolution.

We formulate the complete update process of hidden states as follows,

$$g_i(t) = c \sum_{j \in \mathcal{N}(i)} \alpha_{ij} f\left(h_j(t - \tau_{ij})\right), \qquad (15)$$

$$z_i(t) = \sigma(W_z h_i(t) + U_z g_i(t) + b_z),$$

$$\frac{dh_i(t)}{dt} = (1 - z_i(t)) \odot (g_i(t) - h_i(t))\tilde{f}\left(\frac{dX_i(t)}{dt}\right),$$

$$h_i(t) = \begin{cases} \phi_i(t), & t \leq 0 \\ h_i(0) + \int_0^t \frac{dh_i(t)}{dt}dt, & t > 0. \end{cases}$$

## 4.4 Continuous Output Module

We employ another STDDE to generate the final outputs. In this approach, we consider the last stage of the hidden flow in the input process as the history function for the output process. This strategy offers two key advantages: Firstly, the hidden states remain continuous within the manifold space, ensuring unity between the input and output processes. Secondly, unlike traditional output layers that provide predictions with a fixed horizon, we can accurately predict traffic flow at various frequencies. It provides more flexibility and adaptability to different scenarios.

$$\frac{dh(t)}{dt} = (1 - z(t)) \odot (g(t) - h(t)) \qquad (16)$$

$$h_i(t') = h_i(T) + \int_T^{t'} \frac{dh_i(t)}{dt}dt,$$

$$y_i(t') = f\left(h_i(t')\right),$$

$$Y_i = [y_i(t_{T+1}), y_i(t_{T+2}), \cdots, y_i(t_{T+T'})]$$

where $f$ is a mapping function to get final outputs from hidden states, and $t'$ is any target output point. This output approach better highlights the continuity of STDDE and fully capitalizes on its capabilities. With this model, we have the flexibility to generate predictions at any time, rather than being limited to a specific point. The continuous output module is well-suited for scenarios involving sparse traffic sensor data, especially when a higher level of precision is required during inference than in the training phase.

Finally, as our objective function in the context of traffic flow forecasting, we employ the widely-used Huber loss [16], which is known for its robustness in handling outliers compared to the squared error loss.

$$\mathcal{L}(Y, \hat{Y}) = \begin{cases} \frac{1}{2}(Y - \hat{Y})^2 & , |Y - \hat{Y}| \leq \delta \\ \delta|Y - \hat{Y}| - \frac{1}{2}\delta^2 & , \text{otherwise} \end{cases} \qquad (17)$$

where $\hat{Y}$ is the ground truth, and $\delta$ is a hyperparameter which controls the sensitivity to outliers.

The whole flow of STDDE is presented in Algo 1 in the Appendix.

## 4.5 Why It Works?

*4.5.1 Connection to Existing Works.* STDDE includes both delay effects and continuity into a unified delay differential equation framework, which **explicitly models the time delay in spatial information propagation**. A prior related study on delay-aware traffic forecasting is PDFormer [18]. While PDFormer mentions the concept of time delay, its core mechanism involves utilizing attention with the historical time series, rather than explicitly utilizing delay in the propagation process. It implies that it still cannot capture the delay information in both spatial and temporal views.

Then we analyze the generalizability of STDDE from two perspectives: 1) In the temporal dimension, where GRU and its variants [13] can be considered as special cases of STDDE when integration $\int \frac{dh_i(t)}{dt}$ is discrete. 2) In the spatial dimension, when all the time-delays $\tau_{ij}$ are set to zero, STDDE will degenerate to general GNNs [15, 19].

*4.5.2 Stability.* Stability is a critical property for a DDE. Here we provide a theoretical analysis of our proposed DDE.

**Definition 1.** A delay differential equation is linked to a characteristic equation. If the real parts of all characteristic roots of the associated equation are negative, the delay differential equation is considered asymptotically stable.

**Theorem 1.** *The proposed DDE is asymptotically stable when the balance constant $c \leq 1/K$.*

Proof. Please see the Appendix for more details about the definition and the proof.

# 5 EXPERIMENTS

## 5.1 Datasets

We verify the performance of STDDE on six real-world traffic datasets, namely PeMSD7 (M), PeMSD7 (L), PeMS03, PeMS04, PeMS07, and PeMS08, which are collected by Caltrans Performance Measurement System in real-time every 30 seconds [6] and aggregated into 5-min intervals, which means there are 288 time-steps for one day. More details of the datasets are listed in Table 1. We standardize the input by removing the mean and scaling to unit variance.

| Datasets | #Sensors | #Edges | Time Steps |
|----------|----------|--------|------------|
| PeMSD7 (M) | 228 | 1132 | 12672 |
| PeMSD7 (L) | 1026 | 10150 | 12672 |
| PeMS03 | 358 | 547 | 26208 |
| PeMS04 | 307 | 340 | 16992 |
| PeMS07 | 883 | 866 | 28224 |
| PeMS08 | 170 | 295 | 17856 |

**Table 1: The summary of datasets used in our paper.**

## 5.2 Baselines

We select the following representative baselines as our competitors, and more details can be found in the Appendix:

- **Spatio-Temporal Graph Convolution Models** including STGCN [41], STSGCN [35], DCRNN [25], AGCRN [3], AST-GCN [14], FOGS [31]. STGCN utilizes graph convolution and 1D convolution to capture spatial-temporal correlations. STSGCN utilizes multiple localized spatial-temporal subgraph modules to capture the spatial-temporal correlations directly. DCRNN integrates graph convolution into an encoder-decoder gated recurrent unit. AGCRN captures node-specific spatial and temporal correlations automatically without a pre-defined graph. ASTGCN utilizes spatial and temporal attention mechanisms to model their dynamics. DSTAGN constructs a dynamic graph instead of relying on a pre-defined static one. FOGS utilizes first-order gradients rather than specific flows, which effectively circumvent issues associated with fitting irregularly-shaped distributions.
- **Spatial-Temporal Graph Ordinary Differential Equation Models**, including STG-ODE [12] and STG-NCDE [8]. STGODE proposes an ordinary differential equation-based continuous GNN, to capture long-range spatial-temporal dependencies. STG-NCDE designs two NCDEs to capture temporal and spatial properties respectively.

- **Delay-aware Traffic Models** only include one related work, which is PDFormer [18]. Its transformer-based mechanism involves utilizing attention to the historical time series.

## 5.3 Experimental Settings

We split all datasets with a ratio of 6: 2: 2 into training sets, validation sets, and test sets. One hour of historical data is used to predict traffic conditions in the next 60 minutes, i.e. $T = 12$ and $T' = 12$. All experiments are conducted on the same Linux server and GPU. The dimension of hidden states is set to 64. We train our model using the Adam optimizer with a learning rate of 0.001. The batch size is 32 and the training epoch is 200. Mean absolute error (MAE), mean absolute percentage error (MAPE), and root mean squared error (RMSE) are used to measure the performance. For baselines, we use their officially reported results if accessible. If not, we run their codes based on their recommendation configurations.

## 5.4 Conceptual Experiments

We first provide conceptual experiments to evaluate the necessity of motivation and the effectiveness of the proposed delay-aware differential equation and continuous output module.

*5.4.1 Quantitative and Qualitative Validation of Time-delay* . For quantitative validation, we design an invariant of our model, STDDE-no-delay, which sets all the delays as zero for comparison. We compute the average delay of each node in PEMS04 and choose the first 15% and the last 15% nodes after sorting the average delay value in ascending order, to compare the performances between STDDE and STDDE-no-delay. Table 3 shows the result. We find that the results of the nodes with larger average delay are much worse than that of nodes with small delays (MAPE is not a stable metric because it is susceptible to the small values), which indicates the difficulty of dealing with long-range correlations. And STDDE achieves a larger improvement for the long-delay nodes due to its ability to model delay effects.

For qualitative validation, we provide a case study to evaluate the effectiveness of the proposed STDDE in capturing time delay in traffic flow forecasting, we carry out a case study in the real-world dataset. We select two connected neighbor nodes 196 and 198 from PeMS04 dataset to visualize the STDDE's perception with time delay in traffic flow forecasting. Results are shown in Fig 4, the prediction results of STDDE are remarkably closer to the ground truth than STDDE-no-delay. In addition, there is a huge rise in node 196's traffic waveform in Fig 4 (a), and the result in (b) shows that STDDE-no-delay performs inaccurate feedback while STDDE does not. It further shows STDDE is able to capture and utilize the time delay information in traffic flow forecasting.

*5.4.2 Flexibility Validation of the Continuous-output Module*. To test the flexibility of our model in real-world scenarios, we introduced a more challenging setting. We still have historical 60-min data to predict the traffic flow in the next 60 minutes. However, during the training process, we set the time interval as 10/15/20 minutes, which means the input steps are 6/4/3, and during the inference process, we change the time interval to 5 minutes. This configuration rigorously assesses the model's adaptability. For

| Dataset | Metric | STGCN | DCRNN | ASTGCN(r) | STSGCN | STGODE | AGCRN | STG-NCDE | DSTAGNN | FOGS | PDFormer | **STDDE** |
|---|---|---|---|---|---|---|---|---|---|---|---|---|
| | RMSE | 7.55 | 7.18 | 6.87 | 5.93 | 5.66 | 5.54 | 5.39 | 5.54 | 5.54 | 5.60 | **5.10** |
| PeMSD7(M) | MAE | 4.01 | 3.83 | 3.61 | 3.01 | 2.97 | 2.79 | 2.68 | 2.78 | 2.76 | 2.81 | **2.56** |
| | MAPE | 9.67 | 9.81 | 8.84 | 7.55 | 7.36 | 7.02 | 6.76 | 6.93 | 6.83 | 7.06 | **6.51** |
| | RMSE | 8.28 | 8.33 | 7.64 | 6.88 | 5.98 | 5.92 | 5.76 | 5.98 | 6.04 | 5.90 | **5.63** |
| PeMSD7(L) | MAE | 4.84 | 4.33 | 4.09 | 3.61 | 3.22 | 2.99 | 2.87 | 2.98 | 2.96 | 2.92 | **2.77** |
| | MAPE | 11.76 | 11.41 | 10.25 | 9.13 | 7.94 | 7.59 | 7.31 | 7.50 | 7.48 | 7.54 | **7.26** |
| | RMSE | 30.42 | 30.31 | 29.56 | 29.21 | 27.84 | 28.25 | 27.09 | 27.39 | 24.85 | 25.96 | **24.52** |
| PeMS03 | MAE | 17.55 | 17.99 | 17.34 | 17.48 | 16.50 | 15.98 | 15.57 | 15.62 | 15.06 | **14.95** | 15.03 |
| | MAPE | 17.43 | 18.34 | 17.21 | 16.78 | 16.69 | 15.23 | 15.06 | 14.74 | 15.03 | 15.58 | **14.69** |
| | RMSE | 36.01 | 37.65 | 35.22 | 33.65 | 32.82 | 32.26 | 31.09 | 31.71 | 31.29 | 29.96 | **29.86** |
| PeMS04 | MAE | 22.66 | 24.63 | 22.94 | 21.19 | 20.84 | 19.83 | 19.21 | 19.38 | 19.44 | 18.31 | **18.11** |
| | MAPE | 14.34 | 17.01 | 16.43 | 13.90 | 13.77 | 12.97 | 12.76 | 12.77 | 12.81 | 12.07 | **12.07** |
| | RMSE | 39.34 | 38.61 | 37.87 | 39.03 | 37.54 | 36.55 | 33.84 | 34.88 | 34.09 | 32.80 | **32.59** |
| PeMS07 | MAE | 25.33 | 25.22 | 24.01 | 24.26 | 22.99 | 22.37 | 20.53 | 21.62 | 20.79 | 19.78 | **19.47** |
| | MAPE | 11.21 | 11.82 | 10.73 | 10.21 | 10.14 | 9.12 | 8.80 | 9.24 | 8.75 | 8.54 | **8.49** |
| | RMSE | 27.88 | 27.83 | 26.22 | 26.80 | 25.97 | 25.22 | 24.81 | 25.08 | 25.36 | 24.61 | **23.81** |
| PeMS08 | MAE | 18.11 | 17.46 | 16.64 | 17.13 | 16.81 | 15.95 | 15.45 | 15.85 | 16.10 | 15.66 | **15.12** |
| | MAPE | 11.34 | 11.39 | 10.6 | 10.96 | 10.62 | 10.09 | 9.92 | 9.93 | 9.85 | **9.61** | 9.74 |

**Table 2: Performance comparison of baselines and proposed STGDDE on six popular used real-world traffic datasets.**

| Data | Metric | STDDE-no-delay | STDDE | Gain |
|---|---|---|---|---|
| | RMSE | 16.97 | 16.86 | 0.65% |
| First 15% | MAE | 11.54 | 11.47 | 0.61% |
| | MAPE | 19.71 | 18.37 | 6.80% |
| | RMSE | 37.72 | 34.59 | 8.30% |
| Last 15% | MAE | 25.06 | 23.24 | 7.26% |
| | MAPE | 14.63 | 13.96 | 4.65% |

**Table 3: Performances facing delays of different extent.**

models. Compared to the linear interpolation method, the STDDE output module can model the inherent continuity and generate more accurate predictions.

### 5.5 Overall Performances and Analysis

Table 2 shows the results of the proposed STDDE model and competitive baselines on traffic flow forecasting tasks in six popular used real-world datasets. We conclude with the following findings:

- Our model yields the best performance regarding all the metrics for most datasets, which suggests the effectiveness of our spatial-temporal delay traffic flow forecasting.
- Continuous spatial-temporal neural networks, i.e. STGODE, STG-NCDE, and STDDE, perform better than traditional GNN-based ones, such as popularly used AGCRN, STGCN, DCRNN, and DSTAGNN. It shows the direction of continuous modeling in spatial-temporal traffic flow forecasting is effective and worth gaining more attention.
- The proposed STDDE and PDFormer generally perform better than other continuous spatial-temporal methods, i.e. STGODE and STG-NCDE, which indicates that capturing and utilizing historical delay-related information is necessary and of great significance.
- STDDE gains better performance than PDFormer, which shows the effectiveness of explicit spatial-temporal delay-aware differential equations and continuous modeling.

### 5.6 Model Analysis

*5.6.1 Ablation Studies.* To verify the effectiveness of different modules of STGDDE, we conduct the following ablation experiments on PeMS04 dataset and compare results with its corresponding variants.

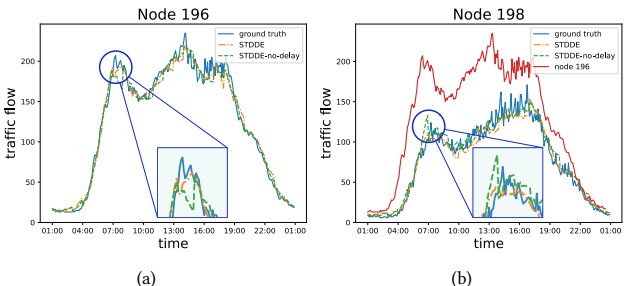

**Figure 4: Comparison of prediction results between our model and STDDE-no-delay.**

STDDE, owing to its continuity, we only need to increase the number of chosen states in the output module, from 6/4/3 to 12. For baseline models, we first acquire their prediction and then adopt the linear interpolation method to acquire a more fine-grained output. The results are presented in Figure 5. In summary, the performance will degrade with the increase of the input interval. The performance of STDDE is significantly better than that of the baseline

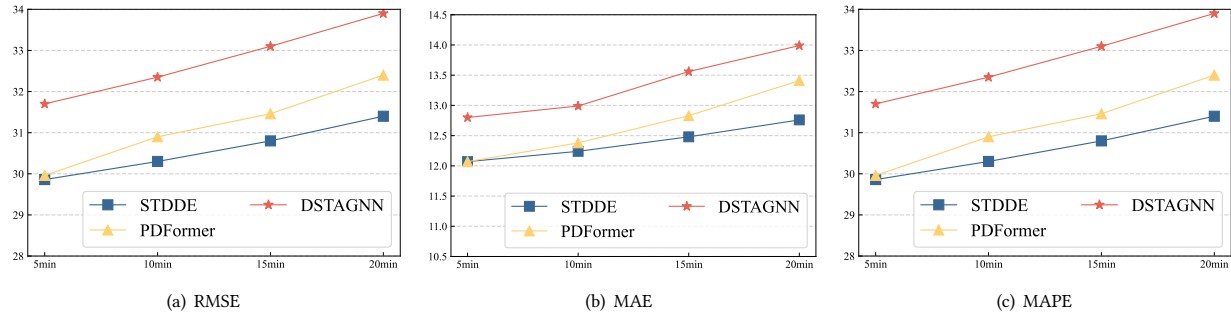

(a) RMSE        (b) MAE        (c) MAPE

**Figure 5: Performance comparison with input time intervals greater than inference intervals.**

| Model | # Parameters | PeMSD7 (M) | | PeMSD7 (L) | |
|---|---|---|---|---|---|
| | | Train | Infer | Train | Infer |
| STGODE | 328,646 | 131 | 13 | 1107 | 146 |
| FOGS | 1,674,188 | 50 | 3 | 531 | 42 |
| DSTAGNN | 2,784,988 | 168 | 43 | 1222 | 209 |
| PDFormer | 531,165 | 120 | 11 | 1292 | 138 |
| STDDE | 175,830 | 82 | 9 | 734 | 84 |

**Table 4: Comparison of # parameters and running time in one epoch. (Unit: seconds)**

- v1 (STDDE-no-delay): In this variant, we take no account of the delay effect, and thus the model degenerates to an ODE model, to verify whether capturing time delay signal is contributing.
- v2 (STDDE-zero-history): We set the history function of STDDE as zero function, to verify the necessity of learnable history states.
- v3 (STDDE-fixed-delay): We use the pre-processed delay values as the inputs of STDDE.

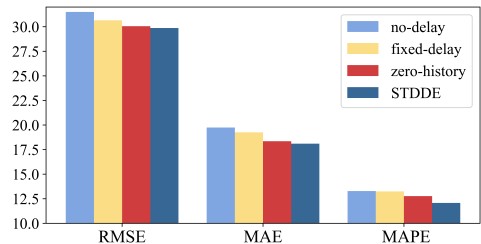

**Figure 6: Ablation experiments of STDDE.**

The result is presented in Fig 6. The result shows that STDDE-no-delay has a significant performance gap with STDDE, which shows the necessity of utilizing time delay. Also, STDDE-fixed-delay performs worse than STDDE, which clearly shows the superiority of learnable delay values. In addition, STDDE performs better than STDDE-zero-history, because the historical states of DDE is critical to the update of a period of future states.

*5.6.2 Model Efficiency Analysis.* We conduct model efficiency analysis on STDDE and several representative baselines, i.e. STGODE, DSTAGNN, FOGS and PDFormer, in PeMSD7 (M) and PeMSD7 (L) datasets. Tab. 4 reports the number of parameters, average training, and inference time per epoch. We find that STDDE achieves competitive computational efficiency in both the training and inference phases. In the largest dataset PeMSD7 (L), compared with the best-performing PDFormer, STDDE reduces the training and inference time by over 40% and 20%, respectively.

*5.6.3 Parameter Analysis.* We analyze the dimension of hidden states in STDDE, which influences the complexity of the state space. Fig 7 shows the result on dataset PEMS04. The performance rises with the increase of hidden dimension and achieves the best when the dimension is 128. Considering the balance of effectiveness and efficiency, we set the dimension as 64 in our model.

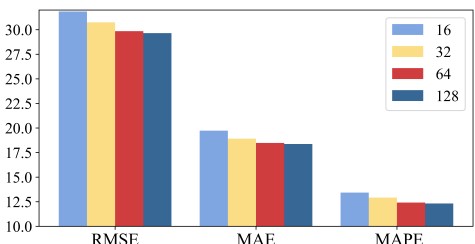

**Figure 7: STDDE results with the change of hidden size.**

## 6 CONCLUSION

In this paper, we propose STDDE which includes both delay effects and continuity into a unified delay differential equation framework. It explicitly models the time delay in spatial information propagation. To gain more accurate depictions of the time delay, we design a traffic-graph time-delay estimator, which provides both precompute delay values and learnable delay modules. In addition, we propose a continuous output module, allowing us to accurately predict traffic flow at various frequencies, which provides more flexibility and adaptability to different scenarios. Finally, we conduct experiments on six popular datasets, in which results show that our model outperforms the SOTAs and exhibits competitive computational efficiency.

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

# 7 APPENDIX

## 7.1 Algorithm Pseudocode

The whole flow of STDDE is presented as follows.

---

**Algorithm 1** STDDE

---

**Require:** Input series $\{X_{t-T+1:t,1}, X_{t-T+1:t,2}, \cdots, X_{t-T+1:t,N}\}$ for all $N$ nodes with series length $T$, graph $G$

**Ensure:** Output series $\{X_{t+1:t+T',1}, X_{t+1:t+T',2}, \cdots, X_{t+1:t+T',N}\}$ for all nodes with series length $T'$

1: Calculate delay $\tau$ for all node pairs through MCC (12) or a learnable estimator;
2: Generate continuous paths $X$ for all nodes through natural cubic spline;
3: Generate the history function $\phi(t)$ based on input data series through Eqn.(10);
4: Generate the hidden states and update the state flow through STDDE (15);
5: Acquire outputs through another STDDE evolution (16);

---

## 7.2 Theoretical Analysis

Here we discuss the stability of the proposed DDE. We begin with introducing the basic definitions and lemmas.

**Definition 2.** The logarithmic norm $\mu$ of a square matrix $A$ is defined as

$$\mu_p(A) = \lim_{\delta \longrightarrow 0^+} \frac{||I + \delta A||_p - 1}{\delta}, \tag{18}$$

where $I$ is the identity matrix of the same dimension as $A$, and $|| * ||_p$ denotes an induced matrix norm.

The logarithmic norm is widely used in differential equation analysis [34, 36], and plays an important role in our analysis.

**Lemma 1.** [36] Let $A = (a_{ij}) \in \mathbb{R}^{n \times n}$, then

$$\mu_1(A) = \max_j \left[ a_{jj} + \sum_{i, i \neq j} |a_{ij}| \right], \tag{19}$$

$$\mu_2(A) = \frac{1}{2} \max_i \left[ \lambda_i(A + A^T) \right], \tag{20}$$

$$\mu_\infty(A) = \max_i \left[ a_{ii} + \sum_{j, j \neq i} |a_{ij}| \right]. \tag{21}$$

From theorem1, we see that $\mu_p(A)$ is easy to calculate for $p = 1, \infty$ or estimate for $p = 2$, which brings convenience for the analysis.

**Definition 3.** A linear multi-delay differential equation is defined as

$$\frac{\mathrm{d}h(t)}{\mathrm{d}t} = A_0 h(t) + \sum_{k=1}^{K} A_k h(t_{\tau_k}), \tag{22}$$

where $h(t) = (h_0(t), h_1(t), \cdots, h_N(t)$ is a vector, $A_0, A_k \in \mathbb{R}^{n \times n}$ are constant matrix, and $h(t_{\tau_k}) = (h_1(t - \tau_{k1}), h_2(t - \tau_{k2}), \cdots, h_N(t - \tau_{kN}))$.

Taking the simplest two-variable delay differential equations as an example,

$$\begin{cases} \dfrac{\mathrm{d}u(t)}{\mathrm{d}t} = a_1 u(t) + b_1 v(t - \tau_2) \\ \dfrac{\mathrm{d}v(t)}{\mathrm{d}t} = a_2 u(t) + b_2 v(t - \tau_1), \end{cases} \tag{23}$$

by letting

$$h(t) = (u(t), v(t))^T, h(t_\tau) = (u(t - \tau_1), v(t - \tau_2))^T, \tag{24}$$

$$A = \begin{pmatrix} a_1 & 0 \\ 0 & a_2 \end{pmatrix}, B = \begin{pmatrix} 0 & b_1 \\ b_2 & 0 \end{pmatrix}, \tag{25}$$

we have

$$\frac{\mathrm{d}h(t)}{\mathrm{d}t} = Ay(t) + By(t_\tau). \tag{26}$$

**Theorem 2.** The proposed DDE is bounded between two linear multi-delay systems.

PROOF. From eq (9), without considering the control signal, we have

$$0 \leq \frac{\mathrm{d}h_i(t)}{\mathrm{d}t} \leq g_i(t) - h_i(t). \tag{27}$$

The lower bound is easy, and we can reformulate the upper bound as

$$g_i(t) - h_i(t) = -h_i(t) + c \sum_{j \in \mathcal{N}_i} \alpha_{ij} h_j(t - \tau_{ij}) \tag{28}$$

Further, let $H(t) = (h_1(t), h_2(t), \cdots, h_n(t))^T$, we have

$$\frac{\mathrm{d}H(t)}{\mathrm{d}t} \leq -IH(t) + \sum_{k=1}^{K} A_k \cdot H(t_{\tau_k}), \tag{29}$$

where $A_k$ is a matrix of which the $i$-th row contains at most one non-zero element of $c\alpha_{ij}$, and $H(t_{\tau_k}) = (h_1(t_{\tau_{k1}}), h_2(t_{\tau_{k2}}), \cdots, h_n(t_{\tau_{kn}}))$ is the re-group of delayed signals. Moreover, $K$ is equal to the maximum of node degrees, because each edge appears only once in these matrices. □

**Definition 4.** The characteristic equation associated with the differential equation (22) is defined as

$$\det \left[ zI - A_0 - \sum_{k=1}^{N} A_K \exp(-zT_k) \right] = 0, \tag{30}$$

where $\exp(-zT_k) = \mathrm{diag}(e^{-z\tau_{k1}}, e^{-z\tau_{k2}}, \cdots, e^{-z\tau_{kN}})$

**Lemma 2.** [37] If the real parts of all characteristic roots of equation (30) are less than zero, then the system is asymptotically stable.

**Lemma 3.** [37] If the condition

$$\mu(A_0) + \sum_{k=1}^{K} ||A_k|| \leq 0 \tag{31}$$

holds, then lemma 2 holds, and the system is asymptotically stable.

**Theorem 3.** The proposed DDE is asymptotically stable when the balance constant $c \leq 1/K$.

PROOF. We prove the result starting from its upper bound and lower bound. The lower bound is clear to satisfy the condition in lemma 3. We focus on the upper bound. We prove the result for norm $p = \infty$, and the result can be generalized to other norms easily. According to the theorem 1, we have

$$\mu_\infty(A_0) = \mu_\infty(-I) = -1, \tag{32}$$

$$||A_k||_\infty = \max_i \sum_{j=1}^{n} |A_{k_{ij}}| = \max_i c\alpha_{ij(k)}. \tag{33}$$

The second equation holds because there are at most one non-zero element in each row of $A_k$, and $c\alpha_{ij(k)}$ denotes the element in row $i$. Due to that $\alpha_{ij}$ is the normalized weight for graph convolution,

we have

$$||A_k||_\infty = \max_i c\alpha_{ij(k)} \le c, \tag{34}$$

and when $c <= 1/K$,

$$\mu(A_0) + \sum_{k=1}^{K} ||A_k|| \le -1 + K \cdot c \le 0, \tag{35}$$

the condition (33) holds. So, the lower bound and upper bound of the proposed DDE are both asymptotically stable, and they reach the same stable point [40]. As a result, the proposed DDE is asymptotically stable.

□

