# OpenReview forum: "Unveiling Delay Effects in Traffic Forecasting: A Perspective from Spatial-Temporal Delay Differential Equations"
_ACM.org/TheWebConf/2024/Conference — TheWebConf24 Oral_

### Official Review · Reviewer_qWYS · 2023-11-16

**Novelty:** 6
**Technical Quality:** 6

**Review:**

The paper introduces a novel approach to address critical challenges in traffic flow forecasting, namely the immediate nature of message passing in Graph Neural Networks (GNNs) and the need for retraining models for different prediction frequencies. The paper's contributions, including the innovative STDDE model that incorporates temporal delay and a continuous output module for improved flexibility, are well-motivated and promising. The theoretical proofs and extensive experimental validation demonstrate the model's superiority in traffic flow forecasting while maintaining computational efficiency. Overall, this paper offers a significant advancement in the field and has the potential to enhance the accuracy and adaptability of traffic flow forecasting models, making it a valuable addition to the conference.

Novelties:
+ The paper's novelty lies in recognizing and addressing the significant but overlooked issue of variable time delays in the propagation of graph signals for spatial-temporal traffic forecasting. Unlike traditional time series models, which assume consistent delays across all nodes and timestamps, this research introduces a dedicated module to model and account for the varying delays among different nodes and timestamps. This innovation is crucial for more accurate predictions in complex traffic scenarios where delays play a significant role in information propagation.
+ The paper's novelty lies in addressing the often-overlooked concept of continuity within the traffic system. Existing methods, such as RNNs and TCNs, typically work with discrete data, limiting their adaptability to different scenarios and requiring retraining for specific demands. Moreover, the paper recognizes the inherent sparsity of traffic data due to limited sensor availability and aims to achieve finer-grained forecasting precision, allowing rapid responses to events like travel planning or traffic emergencies. This approach enhances the flexibility and adaptability of traffic forecasting models by considering both varying prediction requirements and sparse data conditions.

Contributions: This paper presents three significant contributions: firstly, the introduction of the Spatial-Temporal Delay Differential Equation model (STDDE) that explicitly models time delay in spatial information propagation; secondly, the design of a learnable traffic-graph time-delay estimator to enhance accuracy by considering variations in delay effects; and thirdly, the proposal of a continuous output module that provides flexibility in predicting traffic flow at different frequencies without retraining. These contributions collectively improve the accuracy, adaptability, and flexibility of spatial-temporal traffic forecasting models, addressing key challenges in the field.

Strengths:
+ Comprehensive experiments on six datasets, showcasing the model's superiority over state-of-the-art methods in terms of accuracy and computational efficiency. Furthermore, experiments confirm the effectiveness of the delay-aware module and the adaptability of the continuous output module, solidifying the model's practicality in diverse spatial-temporal traffic forecasting scenarios.
+ Clearly identify the challenges of developing the solution. The challenge addressed in the paper is that traditional Delay Differential Equations (DDE) lack the ability to incorporate incoming inputs once network parameters are fixed, leading to the loss of valuable information. To overcome this limitation, the paper introduces a control signal inspired by Neural Continuous Differential Equations (CDE) to integrate incoming inputs and improve information utilization.
+ Motivations behind each technical design and why the solution works are clearly explained.

Weaknesses:
+ Source code seems to be unavailable yet.
+ The approach mentions the handling of peak hours and non-peak hours but the evaluation seems to be not verify this effect.
+ Time/space complexity is briefly mentioned but not clearly discussed in the evaluation or in theory.

Updated: I have read the rebuttal(s).

**Questions:**

Q1: It would be great if the source code will be available.
Q2: The approach mentions the handling of peak hours and non-peak hours but the evaluation seems to be not verify this effect.
Q3: It would be great to discuss time/space complexity in the evaluation and in theory.

**Reviewer Confidence:**

3: The reviewer is confident but not certain that the evaluation is correct

**Scope:**

3: The work is somewhat relevant to the Web and to the track, and is of narrow interest to a sub-community

---

### Official Review · Reviewer_2t6k · 2023-11-25

**Novelty:** 6
**Technical Quality:** 6

**Review:**

This paper considers Traffic flow forecasting problem, which is critical for transportation planning. Current approaches using Graph Neural Networks (GNNs) and Recurrent Neural Networks (RNNs) face challenges  to model the delay effect into the model.

To address this, the proposed Spatial-Temporal Delay Differential Equation model (STDDE) integrates delay effects and continuity, explicitly modeling time delays. It includes a learnable traffic-graph time-delay estimator and a continuous output module for accurate predictions at various frequencies.

Extensive experiments show STDDE's superior performance and computational efficiency, validating the effectiveness of the proposed modules for flexibility across scenarios.

Strong Points:
- The idea is novel and interesting for this application.
- The paper is well structured and easy to read through.
- The experiment comparison are extensive and informative.

**Questions:**

Since the delay effect does not only appear in the traffic domain, should the authors consider other applications like flight delay? This will increase the scope and audience of this work.

**Reviewer Confidence:**

2: The reviewer is willing to defend the evaluation, but it is likely that the reviewer did not understand parts of the paper

**Scope:**

3: The work is somewhat relevant to the Web and to the track, and is of narrow interest to a sub-community

---

### Official Review · Reviewer_sSXF · 2023-11-27

**Novelty:** 5
**Technical Quality:** 5

**Review:**

Summary:
Traffic flow forecasting is crucial for transportation planning. While Graph Neural Networks (GNNs) and Recurrent Neural Networks (RNNs) excel in capturing spatial-temporal correlations, two unresolved issues persist. Firstly, GNNs lack consideration for time delays in spatial interactions among neighboring nodes. Secondly, existing models require retraining for different prediction frequencies, limiting their adaptability. To address two challenges, this paper proposes the Spatial-Temporal Delay Differential Equation model (STDDE). STDDE integrates delay effects and continuity into a unified framework, explicitly modeling time delays in spatial information propagation. The model includes a learnable traffic-graph time-delay estimator for gradient backward processing and a continuous output module for versatile prediction frequencies. Extensive experiments demonstrate the superior performance of STDDE with competitive computational efficiency, validating the concept of a delay-aware module and the effectiveness of the continuous output module.

Pros:
1. The traffic flow prediction problem tackled in this paper is a very important problem in urban computing and smart transportation.

2. It is very interesting and new to me to consider time delay in traffic flow forecasting and explore the impact of time delay on model performance.

3. Six real-world datasets are used to verify the performance of the model, and the experimental results also illustrate the effectiveness of the proposed model.

Cons:
1. The technical innovation of the method proposed in this article is limited. The proposed model only employs NDDE and NCDE to model the effect of time delay, while the time delay-aware GNN model mentioned in the abstract and introduction section has not been explored in depth.

2. As shown in Table 2, the performance improvement of the proposed model is very small compared to the STOA model. The datasets used are also very small, basically the traffic flow on the highway, and the public traffic datasets in the real urban core area have not been used.

3. The time and space complexity analysis of the proposed model should be given. In the efficiency evaluation experiment (Table 4), the experimental results also need to be explained in detail.

4. The source code of the proposed model was not given during the review stage, so the reproducibility is relatively weak.

**Questions:**

Please see the above cons.

**Reviewer Confidence:**

4: The reviewer is certain that the evaluation is correct and very familiar with the relevant literature

**Scope:**

1: The work is irrelevant to the Web

---

### Official Review · Reviewer_GB4G · 2023-11-28

**Novelty:** 5
**Technical Quality:** 6

**Review:**

Strongness:
1. The document provides a detailed exploration of various models and techniques in traffic forecasting, including spatial-temporal graph convolutional networks, delay-aware traffic models, and differential equation models. This thoroughness ensures a high-quality overview of the topic.
2. It integrates complex mathematical formulations and theories, indicating a high level of technical rigor.
3. The inclusion of comparative analyses and experimental settings adds to the practicality and applicability of the study.
4. The document explores a unique angle by focusing on delay effects in traffic forecasting, which is a less commonly addressed aspect in the field.
5. It discusses several novel approaches and models, like the spatial-temporal delay-aware differential equations, indicating original contributions to the field.
6. The work has significant implications for improving accuracy in traffic forecasting, which is vital for urban planning and management.
7. It provides insights that could lead to the development of more sophisticated and accurate traffic prediction tools.

Weakness:
1. The high level of technicality and dense mathematical content might make it less accessible to readers without a strong background in the subject.
2. Some parts of the study might build upon existing theories and models, which could limit the perceived originality of the work.
3. The practical application of these theories may require further testing and validation in real-world scenarios.

**Questions:**

1. Could you elaborate on the choice of parameters used in your models, particularly in the spatial-temporal delay-aware differential equations? How did you determine the optimal settings for these parameters?
2. In your comparative analysis with other models, did you consider variations in environmental factors or different urban layouts? How might these factors affect the applicability of your models in diverse real-world scenarios?
3. Could you discuss the feasibility of implementing your proposed models in real-world traffic management systems? What are the anticipated challenges and how might they be addressed?
4. How scalable are your models in terms of handling large-scale traffic data, especially in densely populated urban areas? Are there any computational limitations that need to be considered?
5. How robust are your models against irregular or unexpected traffic patterns, such as those caused by accidents or road construction?
6. Could you provide more insight into the data sources used for training and testing your models? How do you ensure the quality and representativeness of this data?

**Reviewer Confidence:**

3: The reviewer is confident but not certain that the evaluation is correct

**Scope:**

4: The work is relevant to the Web and to the track, and is of broad interest to the community

---

### Decision · Program_Chairs · 2024-01-22

**Decision:**

Accept (Oral)

**Comment:**

The reviewers and the area chair agree that this is an interesting contribution to the scientific literature. The paper tackles a problem that has received significant attention and improves on the state of the art with an interesting contribution that is explained with theoretical insights. We recommend acceptance.